# Viral load care of HIV-1 infected children and adolescents: A longitudinal study in rural Zimbabwe

Tichaona Mapangisana[1]*, Rhoderick Machekano[1,2], Vinie Kouamou[3], Caroline Maposhere[4], Kathy McCarty[5], Marceline Mudzana[5], Shungu Munyati[4], Junior Mutsvangwa[4], Justen Manasa[6,7], Tinei Shamu[8,9], Mampedi Bogoshi[10], Dennis Israelski[10], David Katzenstein[4,11]

1 Division of Epidemiology and Biostatistics, Faculty of Medicine and Health Sciences, University of Stellenbosch, Cape Town, South Africa, 2 Elizabeth Glaser Pediatric AIDS Foundation, Washington, DC, United States of America, 3 Department of Medicine, University of Zimbabwe, Harare, Zimbabwe, 4 Biomedical Research and Training Institute, Harare, Zimbabwe, 5 Chidamoyo Christian Hospital, Karoi, Zimbabwe, 6 Department of Medical Microbiology, University of Zimbabwe, Harare, Zimbabwe, 7 African Institute for Biomedical Sciences and Technology, Harare, Zimbabwe, 8 Newlands Clinic, Harare, Zimbabwe, 9 Institute of Social and Preventive Medicine, University of Bern, Bern, Switzerland, 10 Gilead Sciences Inc., Foster City, California, United States of America, 11 Department of Medicine, Stanford University School of Medicine, Stanford, California, United States of America

* tichmuller@gmail.com

**Data Availability Statement:** All relevant data are within the manuscript and its Supporting Information files.

## Abstract

### Introduction

Maintaining virologic suppression of children and adolescents on ART in rural communities in sub-Saharan Africa is challenging. We explored switching drug regimens to protease inhibitor (PI) based treatment and reducing nevirapine and zidovudine use in a differentiated community service delivery model in rural Zimbabwe.

### Methods

From 2016 through 2018, we followed 306 children and adolescents on ART in Hurungwe, Zimbabwe at Chidamoyo Christian Hospital, which provides compact ART regimens at 8 dispersed rural community outreach sites. Viral load testing was performed (2016) by Roche and at follow-up (2018) by a point of care viral load assay. Virologic failure was defined as viral load ≥1,000 copies/ml. A logistic regression model which included demographics, treatment regimens and caregiver's characteristics was used to assess risks for virologic failure and loss to follow-up (LTFU).

### Results

At baseline in 2016, 296 of 306 children and adolescents (97%) were on first-line ART, and only 10 were receiving a PI-based regimen. The median age was 12 years (IQR 8–15) and 55% were female. Two hundred and nine (68%) had viral load suppression (<1,000 copies/ml) and 97(32%) were unsuppressed (viral load ≥1000). At follow-up in 2018, 42/306 (14%) were either transferred 23 (7%) or LTFU 17 (6%) and 2 had died. In 2018, of the 264

**Funding:** Gilead Sciences Inc. provided support for this study in the form an Investigator Sponsored Research award to SM and DK (ISR-17-10142) and salaries to DI and MB. The specific roles of these authors are articulated in the 'author contributions' section. The funder critically reviewed the manuscript, but had no other role in study design, data collection and analysis, decision to publish, or preparation of the manuscript.

**Competing interests:** The authors have read the journal's policy and the authors of this manuscript have the following competing interests: DI and MB are paid employees of Gilead Sciences Inc. There are no patents, products in development or marketed products to declare. This does not alter our adherence to PLOS ONE policies on sharing data and materials.

**Abbreviations:** ART, Antiretroviral Therapy; CBART, Community Based ART; HIV, Human Immunodeficiency Virus; LTFU, Lost to follow up; NNRTI, Non-Nucleoside Reverse Transcriptase Inhibitors; NRTI, Nucleoside Reverse Transcriptase Inhibitors; PI, Protease Inhibitors; 3TC, Lamivudine; AZT, Zidovudine; ABC, Abacavir; TDF, Tenofovir disoproxil fumarate; LPV/r, Lopinavir/ritonavir; ATZ/r, Atazanavir/ritonavir; POC, Point of care; VL, Viral load; VL suppression, Viral load suppression; DSD, Differentiated service delivery.

retained in care, 107/264 (41%), had been switched to second-line, ritonavir-boosted PI with abacavir as a new nucleotide analog reverse transcriptase inhibitor (NRTI). Overall viral load suppression increased from 68% in 2016 to 81% in 2018 (P<0.001).

## Conclusion

Viral load testing, and switching to second-line, ritonavir-boosted PI with abacavir significantly increased virologic suppression among HIV-infected children and adolescents in rural Zimbabwe.

## Introduction

Universal testing and initiation of effective antiretroviral therapy (ART) can mitigate disease progression and the onward transmission of HIV [1]. However, sustaining viral load (VL) suppression of HIV-infected children and adolescent in rural communities is challenging [2–4]. Despite roll out of treatment in sub-Saharan Africa (SSA), adolescents (10–19 years) are at higher risk of unsuppressed viral load compared to adults and children, and continue to have high morbidity and mortality [3–5]. In a national study, the Zimbabwe Population Based HIV Impact Assessment study in 2016 identified viral load suppression, defined as viral load <1,000 copies/ml in less than 50% of children and adolescents receiving first-line ART [6]. Children and adolescents on ART in SSA may demonstrate more frequent virologic failure and low level viremia (60–1,000 copies/ml) compared to adults [7–9].

New guidelines in 2016 for first-line public health ART in low-and-middle-income countries (LMICs) recommended task shifting to nurse led, decentralized ART treatment with a non-nucleoside reverse transcriptase inhibitor (NNRTI) combined with two nucleotide reverse transcriptase inhibitors (NRTI) including lamivudine (3TC) [10]. WHO guidelines, which included tenofovir disoproxil fumarate (TDF) combined with lamivudine and efavirenz (EFV) for adults since 2001, did not recommend tenofovir disoproxil fumarate for children and youth (< 35kg) until 2015 [11,12]. Tenofovir disoproxil fumarate for children was constrained by differing criteria as a preferred nucleotide reverse transcriptase inhibitors based on age, weight and sexual maturity (Tanner stages) and cut-offs by WHO, the US-DHHS and PENTA guidelines [13,14].

Access to generic co-formulation of three drug combinations in LMICs have been driven by cost and availability [15]. Thus, adolescents and children weighing > 35kgs were treated with a single tablet regimen of generic tenofovir disoproxil fumarate/lamivudine/efavirenz. Children weighing < 35kgs, based on available formulations, received twice daily zidovudine/lamivudine/nevirapine (NVP) [16,17]. After virologic failure of a first-line regimen, pediatric co-formulations of generic lopinavir/ritonavir (LPV/r) or atazanavir/ritonavir (ATV/r) as a heat stable tablets were combined with a nucleotide reverse transcriptase inhibitor backbone of abacavir (ABC) and lamivudine [18,19]. Protease inhibitors and a new nucleotide reverse transcriptase inhibitor were recommended with switching because of the extensive drug resistance to first-line therapy [20–23].

Challenges in paediatric treatment include stigma, dependence on caregivers, access to youth friendly health services and enhanced adherence counselling [24–26]. Stigma and incomplete adherence may be mitigated through differentiated service delivery (DSD) models of community based ART (CBART) [27,28]. The DSD model at Chidamoyo Christian Hospital (CCH), a rural mission hospital in Hurungwe district, Zimbabwe comprises CBART in

which a team of healthcare workers make bimonthly visits to distant community ART outreach sites delivering drug and obtaining blood samples for viral load monitoring at dispersed outreach sites. Children and adolescents living less than 10 km from the hospital attend a youth friendly bimonthly clinic at CCH. This CBART delivery model effectively reduces the time, transport and costs imposed on patients when individual visits to a pharmacy and clinic site distant from rural homesteads are required [29,30]. We assessed viral load suppression with viral load monitoring and changes in recommended ART regimens in a nurse-led clinic and community outreach treatment program in rural Zimbabwe.

## Methods

### Study design

This was a longitudinal study of HIV infected children and adolescents receiving ART through CCH in north west Zimbabwe. The study included all children and adolescents who had been on ART for more than 2 years through the Chidamoyo program at either rural outreach sites or the CCH.

### HIV viral load testing

An initial viral load test was obtained at routine outreach and hospital clinic visits between May and September 2016 from children and adolescents on ART for more than two years. Whole blood samples were transported to Harare (300 km one-way trip) within 24 hours, and plasma was separated and kept frozen at -20˚C. The Roche COBAS® Ampliprep®/COBAS Taqman48® HIV-1 v 2.0 test was performed, and the results of the quantitative viral load test were returned to providers within one week. Two years later in 2018, a follow-up sample was obtained at enrollment in the CBART clinical trial (NCT03986099). In this follow-up sample, viral load suppression was ascertained by SAMBA-II semi-quantitative testing at CCH [31,32] and standard of Care—Roche assays at Chinhoyi Provincial Hospital.

### Study setting

Zimbabwe is one of the Southern African countries with high level of inequality, economic decline and high prevalence of HIV, particularly in rural areas. Hurungwe district is a predominantly agrarian area in Mashonaland West province with a prevalence of poverty and extreme poverty of 89.1% and 56.1% respectively (http://www.zimstat.co.zw). CCH is an 85-bed hospital providing medical services to over 200,000 people including immunization outreach, with 300 inpatient admissions and 150 deliveries per month. The community based outreach programme provides ART to over 4,000 HIV patients (https://www.youtube.com/watch?v=f7se3-zhndo).

The service delivery model provides bimonthly ART to adults, adolescents, and children at eight rural outreach sites, 22.5 to 47 km by poor gravel roads (mean 32.8 km) from CCH (see Fig 1). Outreach visits to refill prescription ART drugs, offer adherence counselling, capture vital signs, and assess problems are scheduled every two months. Community health workers inform and remind the community ART recipients to attend. A team from CCH including a nurse, pharmacy assistant and counsellor travel to the outreach site bimonthly to provide services to community care groups of 200–400 people living with HIV.

### Statistical analysis

We categorized viral load as suppressed (viral load <1000 copies/ml) or unsuppressed (viral load ≥1000 copies/ml). Age was categorized into three groups: <10 years, 10–15 years and

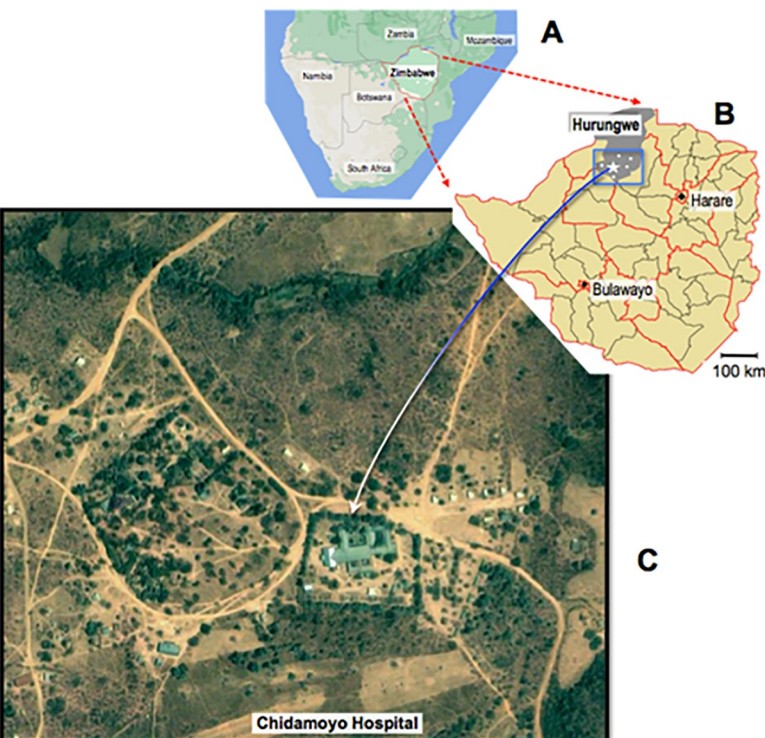

**Fig 1. Hurungwe district in Northwestern Zimbabwe.** A-map of the Southern part of Africa showing Zimbabwe and surrounding countries; B-map of Zimbabwe showing the Hurungwe District in the Northwestern part of Zimbabwe. The approximate location of Chidamoyo Christian Hospitalis indicated by the white star and the relative locations of five outpatient sites (Batanayi, Magororo, Chedope, Nyamutora, and Zvarai) are indicated by white circles; C-the arrow is pointing to Chidamoyo Christian Hospital. This figure was downloaded online from the Humanitarian Response info website (https://www.humanitarianresponse.info/sites/www.humanitarianresponse.info/files/ZWE).

≥15 years. We grouped primary caregivers as both parents, single parent and other non-parent relatives. Information collected included age, gender, weight, clinical and laboratory data (viral load and CD4 count), ART regimen, ART initiation and co-trimoxazole prophylaxis dates, primary caregiver and site of HIV care (hospital versus outreach). We abstracted data from patient treatment register to a structured data retrieval form. We summarized baseline characteristics using frequencies and proportions for categorical variables and medians for continuous variables stratified by baseline virologic outcomes. Chi-square tests, Fisher exact tests and Wilcoxon rank-sum tests were used to assess associations between baseline characteristics and virologic outcomes where appropriate. We compared viral load suppression profile in this cohort between baseline and follow-up using the chi-square test for marginal homogeneity between the paired viral load assessments. Suppression rates for children and adolescents who were switched and those that remained on non-nucleoside reverse transcriptase inhibitor-based first-line therapy after failing were estimated and compared. To understand the role of switching to protease inhibitor-based second-line ART on viral load suppression at follow-up, we fitted a logistic regression model of follow-up viral load status as a function of baseline viral load and an indicator of switching to second-line or not with an interaction term between the two predictors adjusting for age, gender, HIV care center and primary caregiver. Variables associated with viral load suppression at $p < 0.20$ in the univariable analysis and clinically relevant variables were added to the multivariable model. All statistical analyses were performed using Stata 15.1 (College Station, Tx).

### Ethics approval and consent to participate

This study was first approved by the Institutional Review Board (IRB) of the Biomedical Research and Training Institute and then approved by the Medical Research Council of Zimbabwe (MRCZ/A/2269), the national regulatory and ethical board. Assent was obtained from 7 to 17-years old with guardian consent and those aged ≥18 years provided written consent to extract their past (2016) medical record at enrolment into the study.

## Results

### Baseline characteristics, HIV treatment and viral load suppression in 2016

There were 306 children and adolescents on ART with a median (IQR) ART duration of 5.2 years (3.0–6.5) in the Chidamoyo treatment program in 2016, median age (IQR) was 12 years (8–15) and 54% were female. Of the 306 children and adolescents enrolled, 222 (73%) received HIV care at one of the eight outreach sites, while 84 (27%) living within 10 km of CCH received care at the hospital clinic. The overall viral load suppression rate was 68% at baseline. There were no significant associations between, age, gender, caregiver, site of treatment (clinic vs outreach) and viral load suppression (Table 1).

There were 296 (97%) on first-line non-nucleoside reverse transcriptase inhibitor-based regimens and 10 (3%) on second-line protease inhibitor-based regimens; 6 on atazanavir/ritonavir and 4 on lopinavir/ritonavir. First-line non-nucleoside reverse transcriptase inhibitor regimens were either nevirapine (113) or efavirenz (183) plus lamivudine combined with a second nucleotide reverse transcriptase inhibitor, either zidovudine (115) or tenofovir disoproxil fumarate (181). The viral load suppression rate for those receiving efavirenz-based regimens (73%) was significantly higher than the viral load suppression rate of those who were receiving nevirapine-based regimens, (61%, adjusted OR = 2.44, 95% CI: 1.11–5.36, p = 0.026) (Table 1).

### HIV treatment and viral load suppression in 2018

Two years after the baseline survey, 264/306 (86%) were retained in care through the Chidamoyo program; 23 (7%) had transferred to other ART facilities, 17 (6%) were lost to follow-up (LTFU) and 2 died. We evaluated the characteristics of the 42 not included in follow-up and found that only age was significantly associated with transferring or loss to follow up. Participants who were 15 years and older were more likely to transfer their care or lost to follow up compared to other age categories (11 (10%) for <10 years old, 9 (9%) for 10–15 years old and 20 (22%) for adolescents ≥15 years old, p = 0.008).

At follow-up in 2018, 107/264 (40%) had switched to a protease inhibitor-based regimen and 150/264 (57%) remained on a first-line efavirenz-based regimen while 7 (3%) remained on protease inhibitors. Of those on first-line non-nucleoside reverse transcriptase inhibitor-based regimens, 147/150 (98%) were on single daily full dose combination of tenofovir disoproxil fumarate/lamivudine/efavirenz, only 2 children remained on twice daily dosing of zidovudine/lamivudine/nevirapine and 1 was on tenofovir disoproxil fumarate/lamivudine/nevirapine. One hundred and seven had switched to a protease inhibitor-based regimen and 5 remained on their same protease inhibitor-based regimen, either with lopinavir/ritonavir 90/112 (80%) or atazanavir/ritonavir 22/112 (20%). Of the 107 who changed to a protease inhibitor-based regimen, 39 were suppressed on first-line and 68 had a viral load ≥ 1,000 copies/ml (see Fig 2). The 68 with virologic failure who switched to a protease inhibitor and one new nucleotide reverse transcriptase inhibitor had a significantly lower rate of viral load suppression 45/68 (66%) compared to 39 who were switched to a new protease inhibitor-based regimen while suppressed on the non-nucleoside reverse transcriptase inhibitor, 37/39 (95%)

Table 1. Baseline characteristics of children and adolescents stratified by baseline viral load suppression (N = 306).

| Characteristics | | Total sampled | Baseline Viral load (copies/ml) | | Unadjusted Odds Ratio (95% CI) | Adjusted Odds Ratio [95% CI] | p-value |
|---|---|---|---|---|---|---|---|
| | | | Suppressed | Unsuppressed | | | |
| | | | <1000 | ≥1000 | | | |
| N | | 306 | 209(68%) | 97(32%) | | | |
| Age group (years) | <10 | 110 (36%) | 71(65%) | 39(35%) | - | - | |
| | 10–14 | 107 (35%) | 79(74%) | 28(26%) | 1.55 [0.87–2.77] | 0.84 [0.38–1.85] | 0.669 |
| | 15–23 | 89 (29%) | 59(66%) | 30(34%) | 1.08 [0.60–0.94] | 0.52 [0.21–1.28] | 0.153 |
| Gender | Female | 165 (54%) | 117(72%) | 48(28%) | - | | |
| | Male | 141 (46%) | 92(64%) | 49(36%) | 0.77 [0.47–1.25] | 0.80 [0.49–1.30] | 0.368 |
| HIV care center | Hospital | 84 (27%) | 61(73%) | 23(27%) | - | | |
| | Outreach clinics | 222 (73%) | 147(66%) | 74(33%) | 0.75 [0.43–1.31] | 0.77 [0.43–1.36] | 0.367 |
| Caregiver | Both parents | 34 (12%) | 24(71%) | 10(29%) | - | | |
| | Single parent | 142 (51%) | 101(71%) | 41(29%) | 1.03 [0.45–2.34] | | |
| | Non-parent | 105 (37%) | 68(65%) | 37(35%) | 0.77 [0.33–1.77] | | |
| ART line | First-line (NNRTI) | 296(97%) | 202(68%) | 94(32%) | - | | |
| | Second-line (PI) (ATV/r or LPV/r) | 10 (3%) | 7(70%) | 3(30%) | 1.08 [0.27–4.29] | | |
| ART regimen | NVP | 113(37%) | 69(61%) | 44(39%) | - | - | - |
| | EFV | 183(60%) | 133(73%) | 50(27%) | 1.70 [1.03–2.79] | 2.44 [1.11–5.36] | 0.026 |
| | PI (ATV/r or LPV/r) | 10(3%) | 7(70%) | 3(30%) | 1.49 [0.37–6.06] | 2.18 [0.47–10.03] | 0.317 |
| ART duration (years) | Median (IQR) | 5.2 (3.0–6.5) | 5.2 (2.8–6.5) | 5.2 (3.2–6.5) | 0.95 [0.86–1.05] | | |
| CD4 (cells/cu mm) | <500 | 47(31%) | 18(38%) | 29(62%) | - | - | |
| | ≥500 | 106(69%) | 79(75%) | 27(25%) | 4.71 [2.26–9.81] | | |

ART-Antiretroviral therapy, PI-Protease inhibitor, NNRTI-Non-nucleoside reverse transcriptase inhibitor, NVP-Nevirapine, EFV-Efavirenz, IQR-Interquartile range

(p = 0.001). Those with viral load suppression who remained on first-line therapy were more likely to maintain viral load suppression compared to those with viral load > 1,000 copies/ml who switched to second-line: 86% vs 74%, respectively (p = 0.047). The odds of being suppressed were higher in children and adolescents who were on non-nucleoside reverse transcriptase inhibitor-based first-line regimens (and had not previously failed) compared to those who were on protease inhibitor-based second-line therapy (OR = 0.44, 95% CI: 0.22–0.90, p = 0.024). There were no significant differences in viral load suppression rates by age group, caregiver, weight, and gender (Table 2).

## The treatment cascade and virologic outcomes between 2016 and 2018

Among 175 with viral load suppression in 2016 on a first-line non-nucleoside reverse transcriptase inhibitor-based regimen, 136 (78%) remained on a first-line non-nucleoside reverse transcriptase inhibitor-based regimen, mostly as a single tablet regimen combining tenofovir disoproxil fumarate/lamivudine/efavirenz with 122/136 (90%) suppressed in 2018 as shown in

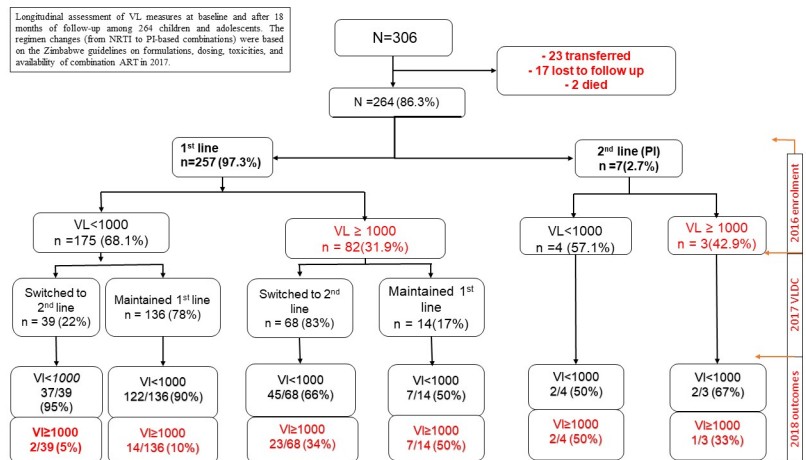

**Fig 2. Longitudinal assessment of viral load measures at baseline and after 18 months of follow-up among 264 children and adolescents.** The regimen changes (from non-nucleoside reverse transcriptase inhibitor to protease inhibitor-based combinations) were based on the Zimbabwe guidelines on formulations, dosing, toxicities, and availability of combination ART in 2017. Virologic failure (≥1,000 copies/ml) at enrollment and follow-up are indicated in red.

Fig 2. Similarly, of the 39 who were suppressed in 2016 and switched to a protease inhibitor-based regimen, 37/39 (95%) were virological suppressed in 2018. Of the 68 unsuppressed children and adolescents on a first-line non-nucleoside reverse transcriptase inhibitor-based regimen, in 2016 who switched to a protease inhibitor, only 45 (66%) achieved virologic suppression compared to 7/14 (50%) among those failing at baseline who remained on first-line ART (p = 0.252). Among the 179 suppressed adolescents and children at baseline, 161 (90%) remained suppressed in 2018. Overall, the proportion of children achieving viral load suppression at follow-up was significantly higher than at baseline (81% vs. 68%, p<0.001).

## Factors associated with viral load suppression of children and adolescents at follow-up

Table 3 presents the adjusted effect of switching to protease inhibitor-based ART on virologic outcomes for children and adolescence who were on first-line non-nucleoside reverse transcriptase inhibitor-based therapy at baseline. Among the children and adolescents who switched, 77% (82/107) were suppressed compared to 86% (129/150) among children who were maintained on a first-line regimen. Adjusting for baseline viral load, age, gender, site of HIV care and caregiver, switching to second-line ART was not significantly associated with viral load suppression at follow-up (adjusted OR = 2.71, 95% CI: 0.99–7.44, p = 0.053). Viral load suppression at follow-up was significantly and independently associated with baseline viral load suppression (adjusted OR = 9.37, 95% CI: 3.62–24. 26) p<0.001). Viral load suppression rates did not differ significantly by age group, gender, HIV care site nor primary caregiver.

## Discussion

Viral load monitoring and drug switching (from a first-line non-nucleoside reverse transcriptase inhibitor-based regimen to a second-line protease inhibitor-based regimen) in a community-based ART clinic and outreach treatment increased viral load suppression among children and adolescents in rural Zimbabwe. We found that, suppression increased from 68%

**Table 2. Characteristics of 264 children and adolescents stratified by viral load suppression at follow up in 2018 (N = 264).**

| 2018–children and adolescents on ART | | Total sampled (264) | Viral load (copies/ml) | | Unadjusted Odds Ratio [95% CI] | Adjusted Odds Ratio [95% CI] | p-value |
|---|---|---|---|---|---|---|---|
| | | | <1000 n = 215 | > = 1000 n = 49 | | | |
| **Age** | <10 years | 70(27%) | 57(81%) | 13(19%) | | - | |
| | 10 to 15 years | 88(33%) | 78(89%) | 10(11%) | 1.78 [0.73–4.34] | 1.37 [0.52–3.60] | 0.528 |
| | >than 15 years | 106(40%) | 80(75%) | 26(25%) | 0.70 [0.33–1.48] | 0.51 [0.22–1.17] | 0.109 |
| **Gender** | Female | 144(55%) | 121 (84%) | 23(16%) | | | 0.329 |
| | Male | 120(45%) | 94(78%) | 26(22%) | 0.69 [0.37–1.28] | 0.73 [0.38–1.38] | |
| **Site** | Chidamoyo site | 134(51%) | 105 (78%) | 29(22%) | | | 0.156 |
| | Outreach site | 130(49%) | 110 (85%) | 20(15%) | 1.52 [0.81–2.85] | 1.60 [0.83–3.09] | |
| **ART line** | NNRTI | 150(57%) | 129 (86%) | 21(14%) | | | 0.024 |
| | PI | 114(43%) | 86(75%) | 28(25%) | 0.50 [0.27–0.94] | 0.44 [0.22–0.90] | |
| **Care giver** | Both parents | 31(13%) | 23(74%) | 8(26%) | | | |
| | Single parent | 127(52%) | 106 (83%) | 21(17%) | 1.76 [0.69–4.45] | | |
| | Not parent | 88(36%) | 71(81%) | 17(19%) | 1.45 [0.55–3.81] | | |
| **CD4* (cells/mm$^3$) at follow-up** | <500 | 45(29%) | 32(71%) | 13(29%) | | | |
| | ≥500 | 111(71%) | 95(86%) | 16(14%) | 2.41 [1.05–5.56] | | |
| **Weight** | Median (IQR) | 33 (26–46) | 34 (26–46) | 34 (25–48) | 1.00 [0.97–1.03] | | |
| **2018 Duration on ART (years)** | Median (IQR) | 7.0 (5.2–8.2) | 7.2 (5.4–8.2) | 6.4 (4.6–7.9) | 1.07 [0.93–1.21] | | |

ART-Antiretroviral therapy; IQR- Interquartile range

*CD4 numbers were obtained for 156/264 (59%)

in 2016 to 81% in 2018 in a population of economically marginalized children and adolescents in an impoverished rural community in Mashonaland West province. Implementing viral load testing in the context of community-based health service delivery was effective in managing ART and sustaining viral load suppression among children and adolescents, with < 1% mortality and 3% LTFU per year. The change in viral load suppression may be explained by access to viral load testing, increasing use of generic single tablet regimens and more effective drug formulations. Nevirapine and zidovudine were phased out and single tablet regimens of tenofovir disoproxil fumarate/lamivudine/efavirenz was frequently used with the transition from the 2013–2016 guidelines [33].

Viral load testing did not uniformly lead to switching to a second-line protease inhibitor-based regimen. Among the 82 children and adolescents with a viral load ≥1000 on first-line ART at baseline, 68/82 (83%) switched to the recommended second-line and only 14 of 82 (17%) did not. However, of the 14 who on continued first-line with adherence counseling, 7/14(50%) re-suppressed on follow-up testing. This is consistent with studies in Zimbabwe, Ethiopia and South Africa, where re-suppression among first-line failures with adherence

**Table 3. Factors associated with viral suppression of children and adolescents at follow-up (N = 257).**

| Characteristics | Viral suppression n/N (%) | Unadjusted Odds Ratio [95%CI] | Adjusted Odds Ratio [95% CI] | p-value |
|---|---|---|---|---|
| Overall | 211/257(82.1%) | | | |
| **Switched to a PI** | | | | |
| No | 129/150 (86%) | 1 | 1 | - |
| Yes | 82/107 (76.6%) | 0.53 [0.28–1.02] | 2.71 [0.99–7.44] | 0.053 |
| **Baseline viral load** | | | | |
| Failure (> = 1000 copies/ml) | 52/82 (63.4%) | 1 | 1 | - |
| Suppressed (<1000 copies/ml) | 159/175(90.9%) | 5.73[2.90–11.35] | 9.37 [3.62–24.26] | <0.001 |
| **Age group (years)** | | | | |
| <10 | 56/69 (81.2%) | 1 | 1 | - |
| 10–14 | 76/86 (88.4%) | 1.76 [0.72–4.31] | 2.37 [0.83–6.77] | 0.106 |
| 15–23 | 79/102 (77.5%) | 0.80 [0.37–1.71] | 1.00 [0.39–2.55] | 0.997 |
| **Gender** | | | | |
| Female | 119/140 (85.0%) | 1 | 1 | - |
| Male | 92/117 (78.6%) | 0.65 [0.34–1.23] | 0.83[0.40–1.70] | 0.606 |
| **HIV care site** | | | | |
| Hospital | 104/131 (79.4%) | 1 | 1 | - |
| Outreach | 107/126 (84.9%) | 1.46 [0.77–2.79] | 1.70 [0.80–3.358] | 0.165 |
| **Caregiver** | | | | |
| Both parents | 23/31 (74.2%) | 1 | 1 | - |
| Single parent | 104/123 (84.6%) | 1.90 [0.74–4.88] | 2.07 [0.71–6.02] | 0.184 |
| Non-parent | 69/85 (81.2%) | 1.50 [0.57–3.96] | 1.86 [0.60–5.72] | 0.280 |

CI- Confidence interval, PI-Protease inhibitor

counseling alone was reported in 30–60% [26,34–36]. These observations provide support for WHO and recent national guidelines which recommend enhanced adherence counseling and a second viral load test>1,000 copies/ml to confirm virologic failure before switching to a protease inhibitor-based or third-line regimen [10].

Switching to more effective ART regimens in impoverished communities includes the economics and logistics of viral load testing and the prompt return of viral load results [10,37,38]. This is even more relevant and critical in rural areas where viral load testing and technology are often unavailable. Distance from the centralized, urban laboratories running the tests is a formidable barrier to rapid response and adds months to the time to switch in case of failure [39]. Challenges such as electricity availability and status of the roads were demonstrated by viral load quantification in rural sites in Zimbabwe in 2016, which required collection of whole blood and immediate transport on cold chain to a central laboratory. Point of care (POC) viral load assays are emerging [39,40] with evidence of clinical utility, and the Ministry of Health and Child Care implemented SAMBA-II [31,32,41] in Zimbabwe including CCH. This allowed the evaluation of its performance and turn-around time at the site. With four units and a trained laboratory technologist, the SAMBA overcame challenges encountered in central laboratory testing by providing accessible viral load testing in near real time, although throughput was limited to no more than 16–20 samples/day. The value of the SAMBA as a POC viral load test was apparent when torrential rains knocked out power for more than 2 weeks and many roads and bridges became impassable. The SAMBA-II, using solar and battery power, could provide results of samples collected at the clinic or community-based ART outreach sites within 24–48 hours.

Despite the advantages of the SAMBA-II POC viral load test and the successful implementation in rural Zimbabwe, sustaining POC viral load testing may be limited by reagent, equipment and information flow through a laboratory supply chain [42,43]. Moreover, the semi-quantitative SAMBA assay reports virologic failure only for samples that exceed 1000 copies/ml [29–31]. This is in contrast to high throughput centralized laboratory testing, which achieves lower limits of detection of <50 copies/ml and provides quantification greater than 3 $\log_{10}$ copies/ml. With the Roche COBAS® Ampliprep®/COBAS® Taqman48® HIV-1 v2.0 assay in 2016, we observed low level viremia (>50 and < 1,000 copies/ml) in only 8% of children and adolescents, more frequently among protease inhibitor recipients. Although the clinical significance of low level viremia is controversial [44,45], the SAMBA II cut-off is consistent with current WHO recommendations for monitoring viral load suppression in LMICs [10].

Studies in LMICs have shown that virologic failure of first-line ART is associated with reverse transcriptase inhibitor resistance [21–23,46]. Switching from a first-line non-nucleoside reverse transcriptase inhibitor-based regimen to a second-line protease inhibitor-based regimen calls for continued administration of lamivudine and substitution of a new nucleotide reverse transcriptase inhibitor in the new regimen [10]. However, in public health ART programs in Africa, drug stockouts and limited drug availability limit strict adherence to these guidelines [47,48]. Nevertheless, children and adolescents who have failed first-line therapy demonstrated favorable suppression rates on switching to a second-line protease inhibitor-based regimen [30,49]. Here, children and adolescents were changed from nucleotide reverse transcriptase inhibitor fixed dose combinations of tenofovir disoproxil fumarate and lamivudine to abacavir and lamivudine in second-line. The response to a new nucleotide reverse transcriptase inhibitor in the presence of the drug resistance mutations M184V and K65R is predicted to be sub-optimal [23]. Nevertheless, the response to lopinavir/ritonavir or atazanavir/ritonavir and 2 nucleotide reverse transcriptase inhibitors is consistent with studies of adults in Africa where nucleotide reverse transcriptase inhibitor resistance has been found to have little impact on second-line protease inhibitor-based regimens [50–54]. A limitation of the study is that drug resistance testing was not performed on patients failing ART [55].

Sustaining long-term ART and viral load testing in rural Africa in children and adolescents requires innovative approaches to DSD to reduce costs and to improve adherence and retention in care [27,28]. These include youth-friendly clinics [56,57] and CBART, in which patients receive drug refills, adherence counseling and monitoring at non-clinic outreach sites [58–60]. Differentiated service delivery and monitoring in the community may provide more cost-effective and less clinic intensive modes of ART delivery for adolescents and children [29,61,62].

## Conclusions

These observations in rural Zimbabwe show that community-based viral load POC testing and second-line ART for children and adolescents in rural communities is feasible and effective.

## Supporting information

**S1 Data. Data file supporting results of the analyses reported in this study.**
(XLSX)

## Author Contributions

**Conceptualization:** Tichaona Mapangisana, Rhoderick Machekano, David Katzenstein.

**Data curation:** Tichaona Mapangisana, Rhoderick Machekano.

**Formal analysis:** Tichaona Mapangisana, Rhoderick Machekano.

**Funding acquisition:** Mampedi Bogoshi, Dennis Israelski.

**Investigation:** Vinie Kouamou, Caroline Maposhere, Justen Manasa, Tinei Shamu.

**Methodology:** Rhoderick Machekano, David Katzenstein.

**Project administration:** Kathy McCarty, Shungu Munyati, Junior Mutsvangwa.

**Resources:** Kathy McCarty, Shungu Munyati, David Katzenstein.

**Software:** Tichaona Mapangisana, Rhoderick Machekano.

**Supervision:** Kathy McCarty, Shungu Munyati.

**Validation:** Junior Mutsvangwa, Justen Manasa.

**Visualization:** Marceline Mudzana.

**Writing – original draft:** Tichaona Mapangisana, Rhoderick Machekano, Vinie Kouamou, David Katzenstein.

**Writing – review & editing:** Mampedi Bogoshi, Dennis Israelski.

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
