## [Decision Letter · Decision Letter 0]

17 Aug 2020

PONE-D-20-20773

Viral load-differentiated care of HIV-1 infected children and adolescents – a prospective longitudinal study in rural Zimbabwe.

PLOS ONE

Dear Dr. Mapangisana,

Thank you for submitting your manuscript to PLOS ONE. After careful consideration, we feel that it has merit but does not fully meet PLOS ONE’s publication criteria as it currently stands. Therefore, we invite you to submit a revised version of the manuscript that addresses the points raised during the review process.

We look forward to receiving your revised manuscript.

Kind regards,

Joseph Fokam, Ph.D

Academic Editor

PLOS ONE

Journal Requirements:

2. We note that Figure 1 in your submission contain map images which may be copyrighted. All PLOS content is published under the Creative Commons Attribution License (CC BY 4.0), which means that the manuscript, images, and Supporting Information files will be freely available online, and any third party is permitted to access, download, copy, distribute, and use these materials in any way, even commercially, with proper attribution. For these reasons, we cannot publish previously copyrighted maps or satellite images created using proprietary data, such as Google software (Google Maps, Street View, and Earth). For more information, see our copyright guidelines: http://journals.plos.org/plosone/s/licenses-and-copyright.

2.1. You may seek permission from the original copyright holder of Figure 1 to publish the content specifically under the CC BY 4.0 license. 

2.2. If you are unable to obtain permission from the original copyright holder to publish these figures under the CC BY 4.0 license or if the copyright holder’s requirements are incompatible with the CC BY 4.0 license, please either i) remove the figure or ii) supply a replacement figure that complies with the CC BY 4.0 license. Please check copyright information on all replacement figures and update the figure caption with source information. If applicable, please specify in the figure caption text when a figure is similar but not identical to the original image and is therefore for illustrative purposes only.

"Funding for the study was provided as Investigator Sponsored Research (ISR-17-10142) from Gilead Sciences Inc., Foster City Ca. TM and RM received support from the Fogarty International Centre of the National Institutes of Health (D43TW010547). TM was supported by DELTAS Africa Initiative SSACAB (Grant No. 107754/Z/15/Z) of the African  Academy of Sciences (AAS) Alliance for Accelerating Excellence in Science in Africa (AESA), New Partnership for Africa’s Development Planning and Coordinating Agency (NEPAD Agency) and the Welcome Trust (Grant No. 107754/Z/15/Z) and the UK government.

The views expressed in this publication are those of the authors and not necessarily those of other parties mentioned in this publication."

We note that one or more of the authors have an affiliation to the commercial funders of this research study : Gilead Sciences Inc.

3.1. Please provide an amended Funding Statement declaring this commercial affiliation, as well as a statement regarding the Role of Funders in your study. If the funding organization did not play a role in the study design, data collection and analysis, decision to publish, or preparation of the manuscript and only provided financial support in the form of authors' salaries and/or research materials, please review your statements relating to the author contributions, and ensure you have specifically and accurately indicated the role(s) that these authors had in your study. You can update author roles in the Author Contributions section of the online submission form.

3.2. Please also provide an updated Competing Interests Statement declaring this commercial affiliation along with any other relevant declarations relating to employment, consultancy, patents, products in development, or marketed products, etc.  

Please respond by return email with an updated Funding Statement and Competing Interests Statement and we will change the online submission form on your behalf.

Reviewers' comments:

Reviewer's Responses to Questions

**Comments to the Author**

1. Is the manuscript technically sound, and do the data support the conclusions?

Reviewer #1: Partly

Reviewer #2: Yes

2. Has the statistical analysis been performed appropriately and rigorously? 

Reviewer #1: Yes

Reviewer #2: Yes

3. Have the authors made all data underlying the findings in their manuscript fully available?

Reviewer #1: No

Reviewer #2: Yes

4. Is the manuscript presented in an intelligible fashion and written in standard English?

Reviewer #1: Yes

Reviewer #2: Yes

5. Review Comments to the Author

Reviewer #1: The review has been uploaded in pdf format to ORCID for PLOS ONE.

The review has been uploaded in pdf format to ORCID for PLOS ONE.

The review has been uploaded in pdf format to ORCID for PLOS ONE.

The review has been uploaded in pdf format to ORCID for PLOS ONE.

Reviewer #2: The manuscript by Mapangisana and colleagues reports on a prospective study of ART treatment implementation in rural Zimbabwe. The following comments are offered for their consideration.

-This is an effective, well done study of a reasonably large cohort of youth, who are known to face challenges with long term treatment.

-The authors should carefully review Table I and Figure 2 for consistency and agreement. For example, Table I seems to list 202 participants with baseline VL suppression who were on first line NNRTI while Figure 2 states the number at 175.

-Line 168: it is of interest for the reader to know how well attended were the bimonthly counselling and outreach sessions

-Forty-two baseline participants were not in the follow-up cohort. Were their characteristics similar or different from the follow-up cohort?

-Viral load testing did, indeed, appear to be beneficial for care. However, it was not used as the sole criteria for switching, as some baseline VL suppressors switched, and some baseline VL failures stayed on their original first line assignment. The authors might consider making this point in the discussion.

-It appears that this cohort experience emphasizes the concept of a ’poor adherence phenotype’ (baseline VL failure with substantial post-switch VL failure) and a ‘good adherence phenotype’ (with baseline VL suppression, likely to remain suppressed no matter whether maintaining first line or switching).

-Table 2/CD4: while the numbers listed are likely accurate, it is not clear that they are what the reader might be interested in. One would like to know how the CD4 counts were distributed in those with VL suppression s VL failure. It appears that not all subjects have CD4 data available. For example, among the 215 VL suppressors, there are only 127 listed with values for CD4. This can be somewhat confusing and bears clarification.

6. PLOS authors have the option to publish the peer review history of their article (what does this mean?). If published, this will include your full peer review and any attached files.

Reviewer #1: **Yes: **Hartmut M. Hanauske-Abel

Pediatrics; Microbiology, Biochemistry & Molecular Genetics; Obstetrics, Gynecology & Women's Health

Rutgers NJMS

Reviewer #2: No

---

## [Author Response · Author response to Decision Letter 0]

13 Oct 2020

Response to reviewers’ comments

https://journals.plos.org/plosone/s/file?id=wjVg/PLOSOne_formatting_sample_main_body.pdf andhttps://journals.plos.org/plosone/s/file?id=ba62/PLOSOne_formatting_sample_title_authors_affiliations.pdf

Response: The authors have ensured that the manuscript meet PLOS ONE requirements

2. We note that Figure 1 in your submission contain map images which may be copyrighted. All PLOS content is published under the Creative Commons Attribution License (CC BY 4.0), which means that the manuscript, images, and Supporting Information files will be freely available online, and any third party is permitted to access, download, copy, distribute, and use these materials in any way, even commercially, with proper attribution. For these reasons, we cannot publish previously copyrighted maps or satellite images created using proprietary data, such as Google software (Google Maps, Street View, and Earth). For more information, see our copyright guidelines: http://journals.plos.org/plosone/s/licenses-and-copyright. We require you to either (1) present written permission from the copyright holder to publish these figures specifically under the CC BY 4.0 license, or (2) remove the figures from your submission:

Response: Figure 1 was constructed from available caption/map from the Ministry of Health Zimbabwe database. Hence, it is similar but not identical to the original image as it contains the specific study sites inserted by the authors. The authors have specified this in the Figure 1 caption. 

"Funding for the study was provided as Investigator Sponsored Research (ISR-17-10142) from Gilead Sciences Inc., Foster City Ca. TM and RM received support from the Fogarty International Centre of the National Institutes of Health (D43TW010547). TM was supported by DELTAS Africa Initiative SSACAB (Grant No. 107754/Z/15/Z) of the African Academy of Sciences (AAS) Alliance for Accelerating Excellence in Science in Africa (AESA), New Partnership for Africa’s Development Planning and Coordinating Agency (NEPAD Agency) and the Welcome Trust (Grant No. 107754/Z/15/Z) and the UK government. The views expressed in this publication are those of the authors and not necessarily those of other parties mentioned in this publication." We note that one or more of the authors have an affiliation to the commercial funders of this research study: Gilead Sciences Inc.

3.1. Please provide an amended Funding Statement declaring this commercial affiliation, as well as a statement regarding the Role of Funders in your study. If the funding organization did not play a role in the study design, data collection and analysis, decision to publish, or preparation of the manuscript and only provided financial support for research materials. in the form of authors' salaries and/or research materials, please review your statements relating to the author contributions, and ensure you have specifically and accurately indicated the role(s) that these authors had in your study. You can update author roles in the Author Contributions section of the online submission form. Please also include the following statement within your amended Funding Statement. “The funder provided support in the form of material support for salaries for authors [insert relevant initials] but did not have any additional role in the study design, data collection and analysis, decision to publish, or preparation of the manuscript. The specific roles of these authors are articulated in the ‘author contributions’ section.” If your commercial affiliation did play a role in your study, please state, and explain this role within your updated Funding Statement.

Response: The authors have modified the funding statement to "Funding for the study to SM and DK was provided as Investigator Sponsored Research (ISR-17-10142) from Gilead Sciences Inc., Foster City Ca. Gilead Sciences which did not play a role in the study design, data collection and analysis or decision to publish. DI and MB are employees of Gilead Science and critically reviewed the manuscript but did not play a role in the decision to publish. TM and RM received support from the Fogarty International Centre of the National Institutes of Health (D43TW010547). TM was supported (research materials plus salary) by DELTAS Africa Initiative SSACAB (Grant No. 107754/Z/15/Z) of the African Academy of Sciences (AAS) Alliance for Accelerating Excellence in Science in Africa (AESA), New Partnership for Africa’s Development Planning and Coordinating Agency (NEPAD Agency) and the Welcome Trust (Grant No. 107754/Z/15/Z) and the UK government, See highlighted in manuscript lines 394-405.

3.2. Please also provide an updated Competing Interests Statement declaring this commercial affiliation along with any other relevant declarations relating to employment, consultancy, patents, products in development, or marketed products, etc. Within your Competing Interests Statement, please confirm that this commercial affiliation does not alter your adherence to all PLOS ONE policies on sharing data and materials by including the following statement: "This does not alter our adherence to PLOS ONE policies on sharing data and materials.” (as detailed online in our guide for authors http://journals.plos.org/plosone/s/competing-interests). If this adherence statement is not accurate and there are restrictions on sharing of data and/or materials, please state these. Please note that we cannot proceed with consideration of your article until this information has been declared.

Response: The authors have provided an updated Competing Interests Statement declaring this commercial affiliation along with any other relevant declarations relating to employment, consultancy, patents, products in development, or marketed products, etc. The authors also provided an amended Funding Statement declaring this commercial affiliation, as well as a statement regarding the Role of Funders in the study on the online submission forms. DI and MB are employees of Gilead sci. They have no competing interest with respect to consultancy, patents or products marketed or in development and there are no restrictions on sharing of data, all data from the study is freely available.

Reviewer #2 responses

1. The authors should carefully review Table I and Figure 2 for consistency and agreement. For example, Table I seems to list 202 participants with baseline VL suppression who were on first line NNRTI while Figure 2 states the number at 175.

Response: Table 1 gives information on the 306 participants who were enrolled in this study. Of the 306 participants, 202 were suppressed on first line regimens. Figure 2 describes these 202 participants who were eventually followed in 2018. Thus, out of the 264 participants who were followed, 175 (68.1%) were suppressed on first line ART regimens.

2. Line 168: it is of interest for the reader to know how well attended were the bimonthly counselling and outreach sessions

Response: The bimonthly counselling and outreach sessions were amazingly well attended by most of the participants and their care givers. 

3. Forty-two baseline participants were not in the follow-up cohort. Were their characteristics similar or different from the follow-up cohort?

Response: A revision has been made to line 229-235. We evaluated the characteristics of the 42 participants who were not included in the follow-up and found that only age was significantly associated with transferring or loss to follow up. Participants who were at least 15 years of age were more likely to transfer their care or lost to follow up.

4. Viral load testing did, indeed, appear to be beneficial for care. However, it was not used as the sole criteria for switching, as some baseline VL suppressors switched, and some baseline VL failures stayed on their original first line assignment. The authors might consider making this point in the discussion.

Response: The authors agree with the reviewer that the viral load testing was not the sole criteria for switching. In the discussion (lines 308-316), we note that PI switching for virologic failure was not uniform although most (83%) of first line failures were switched and only 17% were not.

5. It appears that this cohort experience emphasizes the concept of a ’poor adherence phenotype’ (baseline VL failure with substantial post-switch VL failure) and a ‘good adherence phenotype’ (with baseline VL suppression, likely to remain suppressed no matter whether maintaining first line or switching).

Response: We agree that baseline VL suppression predicted follow-up suppression no matter whether maintaining first line or switching. However we did not capture individual adherence measures, although adherence counselling was provided at each 2 monthly visit as drugs were dispensed, Thus as the reviewer notes, we found that baseline viral load suppression is the most significant, independent predictor of VL suppression at follow-up as shown in Table 3 and in line 280-289. 

6. Table 2/CD4: while the numbers listed are likely accurate, it is not clear that they are what the reader might be interested in. One would like to know how the CD4 counts were distributed in those with VL suppression s VL failure. It appears that not all subjects have CD4 data available. For example, among the 215 VL suppressors, there are only 127 listed with values for CD4. This can be somewhat confusing and bears clarification.

Response: We agree with the reviewer that this is confusing. In fact, 41% of the CD4’s for participants were missing. Thus, this variable was excluded from the multivariate model. We have indicated missingness as a footnote to Table 2. 

1. It is disturbing to see the authors classifying themselves into two separate groups that, though apart for unspecified reasons, nevertheless state each had contributed equally to the manuscript: Are the unspecified contributions of the first-named group (line 27) ‘separate but equal’ to those of the second-named &group (line 28) and therefore identical with the once legal principle of racism and discrimination in the United States ? That principle has been discredited. Or is the first-named group claiming to be ‘more equal’ than the second-named & group, reminiscent of Orwell’s Animal Farm ? Disturbingly, not a single one of the authors in the first-named group states that their primary affiliation is anchored at an institution in Zimbabwe. In the second-named &group of 10 authors, by contrast, those with Zimbabwean names and / or institutional affiliations predominate by a ratio of 9/1. This separation into two classes of authors is evident on the title page. The authors must find a way to put an end to such apartheid. I suggest they follow PLOS convention and specify select categories of contributions, e.g. similar to the ones in ref. [2]:

Response: The authors have rearranged and specified the authors contributions as per PLOS one guidelines. Number 1 being the main author and other co-authors. The authors contribution was summarised as in table below:

Table 1: Authors contributions

Contributor Role Authors

Conceptualization TM, DK, RM 

Data Curation TM, RM

Formal Analysis TM RM

Funding Acquisition DI, MB

Investigation CM, TS, VK, JM6,7 

Methodology DK, RM

Project Administration KM, SM, JR M 

Resources KM, DK, SM, 

Software TM RM

Supervision SM, KM

Validation JM6,7, JM4

Visualization MM

Writing – Original Draft Preparation DK, TM, VK, RM

Writing – Review & Editing DI, MB

JM6,7 - Justen Manasa , JM4- Junior Mutsvangwa

2. The title of PONE-D-20-20773 specifically states that this is a “prospective longitudinal study” that ran from 2016 to 2018. On close reading, in particular of the Methods section on ‘HIV VL testing’, it appears that the 2018 segment of the study rests entirely on data obtained as part of CBART (Community Based Virus Load Differentiated Care in Rural Africa). But CBART did not exist in 2016, the initial year of the findings reported in PONE-D-20-20773. How then might it be possible that a two-year “prospective longitudinal study” is IRB-approved in 2016, at its inception, yet uses clinical protocol and data of CBART, which started years later, on February 1, 2018. From a 2016 IRB perspective, CBART represents an extrinsic and later protocol. This fact, in turn, might raise the added complication of post-hoc analysis of data, up to and including the possible prospect that PONE-D-20-20773 itself could, from its 2016 start, not be a strictly true and genuine “prospective longitudinal study”.

 The Methods section on ‘Ethics approval and consent to participate’ does neither address nor resolve the conundrum of how the 2018 CBART dovetails with the 2016 IRB approval of PONE-D-20-20773. Rather, the formal introduction of two different IRB approvals - “by the Medical Research council of Zimbabwe (MRCZ/A/2269) and by the Biomedical Research Training Institute IRB (AP143/2018)” – further enhances this conundrum: Where participants enrolled in MRCZ/A/2269 and in AP143/2018 ? 

Did all the patients / parents sign two different assent /consent forms? 

No only the consent form signed is for the MRCZ approved study

An attentive reader is left with the impression that PONE-D-20-20773 might have been skilfully tiled and assembled from different protocols

If this should be so, this fundamental ethics issue might be resolved properly by, and might require, an IRB approval for human data use. 

To address this matter objectively, I suggest that the currently unavailable and inaccessible IRB approvals MRCZ/A/2269 and AP143/2018 are uploaded to the PLOS ONE submission site for review. They are not disclosed at this time, just cite

Such an IRB approval for ‘data tiling and compiling’ will leave another conundrum unresolved, however: The CBART study, which supposedly completed in February 2020, specifically defines the identification of “drug resistance mutations” as one of its two Secondary Outcome Measures, the other one being the “Number of participants with confirmed virology failure who switched regimens. Change to second line regimen after confirmed virological failure” (see: https://clinicaltrials.gov/ct2/show/NCT03986099). Yet PONE-D-20-20773, which rests on that very CBART study, states “that drug resistance testing was not performed on patients failing ART” (line 351). The authors themselves instantly recognize this to be “a limitation of the study” (line 350). The authors leave utterly unmentioned that mutation analysis is integral to CBART. This raises questions: Why do they not use CBART data on “drug resistance mutations”, available at least per protocol, in a manner that renders PONE-D-20-20773 as strong scientifically as it could and should be? Are certain CBART data too good for PONE-D-20-20773? Is it really the authors’ choice to submit to PLOS ONE a manuscript for publication that, in their words, is weakened by their own choosing? The authors must resolve these conundrums.

Response: The authors agree with the reviewer and apologize for the word prospective. The CBART Study indeed started in 2018 and assent/consent was obtained from the participants to extract their past (2016) clinical records. The authors have removed the word Prospective in the study design and have revised the ethical approval section for better clarity (See highlighted line 201-205). This specific paper focuses on the baseline Viral load (2016 Baseline viral load) of the enrolled participants, data on objectives of the CBART study are still being curated.

The IRB approval from the BRTI -AP143/2018 is an institutional review that precedes the consent and enrolment form MRCZ/A/2269, the national regulatory and ethical IRB which approved the study. The study was first approved by the institution before it proceeds to the national regulatory and ethical board.

The conundrum is an important question, with a simple answer. The CBART study is a randomized trial of POC and SOC virus load testing, with the secondary endpoints (rate of drug switching and resistance testing). PONE-D-20-20773 includes only the CBART enrolment virus load data obtained between Feb 2018-and September 2018. The full CBART study (see: https://clinicaltrials.gov/ct2/show/NCT03986099) includes resistance testing of the prospectively enrolled and followed “confirmed virologic failures” as noted by the reviewer. The period in the longitudinal study PONE-D-20-20773, did not include resistance testing and this is recognized as limitation of the longitudinal study (lines 356-357). “A limitation of the study is that drug resistance testing was not performed on patients failing ART”

3. Per the WHO HIV Treatment Guidelines and academic publications (see [2]), a core aspect of viral load-differentiated care is to “both serve the needs of PLHIV (people living with HIV) better and reduce unnecessary burdens on the health system” [1], to “reduce the frequency of clinic visits for patients stable on ART” with “anticipated reductions in the costs of clinic visits, due to these being less frequent for many patients”[2]. Consequently, “differentiated care models could decrease health systems costs in 38 countries in sub-Saharan Africa” [3]. This decisive aspect of viral load-differentiated care is entirely absent from PONE-D-20-20773, and its focus on just viral load management makes the study almost a description of routine clinical decisions – clearly, every doctor entrusted with the care of HIV patients will adjust the ARV regimen in response to viral load. By focusing on viral load-differentiated care, the authors have given themselves the burden of showing more than just viral load responses, and they have to answer simple questions like ‘Did the patients who responded to viral load-differentiated care in actual fact require fewer clinic visits, fewer medical resources, and did they have fewer days lost at school, fewer hours lost on the way to clinic, fewer hours wasted while waiting to be seen by the medical team ? ‘If the authors can create a clinic-based quantitative monetary parameter, even better (e.g. costs, ‘how many bandages used? ’, etc.)

I respectfully submit it might be possible to read some of the answers to such questions, essential to viral load-differentiated care, off the medical charts of enrolled patients. The inclusion of such data will turn this manuscript into a reality-based contribution to the discussion on resource utilization in viral load-differentiated care. I encourage the authors to make that effort.

Response: The authors agree with the reviewer, this paper focuses on the 2016 baseline characteristics and follow-up VL of the participants enrolled in the CBART protocol in 2018. For PONE-D-20-20773, the authors have removed the word “differentiated” VL throughout the manuscript as the management of participants was ecological and treatment decisions were made by nurse-providers at Chidamoyo Hospital. 

4. I note a number of the usual diligence issues that reviewers like to discover as evidence of their own diligence, such as spelling and capitalization (e.g. line 10: ‘Cape town’), bringing references up to date and into PLOS format (e.g. ref. 12 is one of many that lack the specified format), improving the pixilation of images (e.g. Fig. 1 lacks the proper resolution), and clarifying graphics (e.g. Fig. 2: Why were patients with VL < 1000 switched to 2nd line ? Why were patients with VL ≥ 1000 maintained on 1st line?).

Response: Cape Town line 10 was corrected. All the references including reference 12 were revised and updated according to PLOS ONE reference format. The pixilation of images was also improved. With regard to figure 2, the changes in regimens were identified in 2017 through chart review. We have attached a figure legend which includes:

Longitudinal assessment of VL measures at baseline and after 18 months of follow-up among 264 children and adolescents. The regimen changes (from NRTI to PI-based combinations) were based on the Zimbabwe guidelines on formulations, dosing, toxicities, and availability of combination ART in 2017.

---

## [Decision Letter · Decision Letter 1]

3 Dec 2020

PONE-D-20-20773R1

Viral load care of HIV-1 infected children and adolescents: a longitudinal study in rural Zimbabwe.

PLOS ONE

Dear Dr. Mapangisana,

Thank you for submitting your manuscript to PLOS ONE. After careful consideration, we feel that it has merit but does not fully meet PLOS ONE’s publication criteria as it currently stands. Therefore, we invite you to submit a revised version of the manuscript that addresses the points raised during the review process.

We look forward to receiving your revised manuscript.

Kind regards,

Joseph Fokam, Ph.D

Academic Editor

PLOS ONE

Reviewers' comments:

Reviewer's Responses to Questions

**Comments to the Author**

1. If the authors have adequately addressed your comments raised in a previous round of review and you feel that this manuscript is now acceptable for publication, you may indicate that here to bypass the “Comments to the Author” section, enter your conflict of interest statement in the “Confidential to Editor” section, and submit your "Accept" recommendation.

Reviewer #1: (No Response)

Reviewer #2: All comments have been addressed

2. Is the manuscript technically sound, and do the data support the conclusions?

Reviewer #1: Partly

Reviewer #2: Yes

3. Has the statistical analysis been performed appropriately and rigorously? 

Reviewer #1: Yes

Reviewer #2: Yes

4. Have the authors made all data underlying the findings in their manuscript fully available?

Reviewer #1: Yes

Reviewer #2: Yes

5. Is the manuscript presented in an intelligible fashion and written in standard English?

Reviewer #1: Yes

Reviewer #2: Yes

6. Review Comments to the Author

Reviewer #1: I have given a detail comment in the pdf file.

Please see the uploaded pdf file

Please see the uploaded pdf file

Reviewer #2: (No Response)

7. PLOS authors have the option to publish the peer review history of their article (what does this mean?). If published, this will include your full peer review and any attached files.

Reviewer #1: **Yes: **Hartmut M. Hanauske-Abel, MD PhD

Reviewer #2: No

---

## [Author Response · Author response to Decision Letter 1]

13 Dec 2020

PONE-D-20-20773-R1 

Second review of Tichaona Mapangisana et al. 

Viral load care of HIV-1 infected children and adolescents: a longitudinal study in rural Zimbabwe. 

The authors’ revisions are sincerely appreciated. 

Before proceeding, I wish to respectfully submit to the authors to kindly consider that with their decision to submit to PLOS ONE, their paper is no longer directed to a specialty audience, as was their presentation to the 11th International Workshop on HIV Pediatrics in July last year on which their manuscript is based (Abstract #51, http://regist2.virology-education.com/abstractbook/2019/abstractbook_Pediatrics2019.pdf ). They now have elected to place themselves onto a different and much elevated stage. In order to earn the full impact that this stage offers and that their effort deserves, they need to leave the previous small format and its specialty language well behind, they need to raise their voice, need to argue additional context and attract the attention of a much wider public with clarity, diligence, and precision. I encourage them to re-examine their manuscript from the much more global perspective of a much more global audience. In that contact, I wish to advance the following suggestions for their consideration: 

1. Foremost, the authors must put the Chidamoyo Hospital data documenting “suppression increased from 68% in 2016 to 81% in 2018 ” into a context that a non-African and non-HIV specialist audience understands and can appreciate. Writing “Whole blood 154 samples were transported to Harare (300 km one-way trip)” is just not enough and invites imagery of car rides on autobahn and interstate, at least 

of driving on asphalted roads ... Not so. 

The authors MUST detail the setting of that hospital - the next major asphalted road is hours away, the remote rural population it serves is economically so disenfranchised that wearing of soled leather shoes is unheard of, per capita income is even less than the $2000 annual that 50% of Zimbabweans earn (https://www.averagesalarysurvey.com/zimbabwe, https://en.wikipedia.org/wiki/Economy_of_Zimbabwe). In fact, just a few months ago one of the authors was forced at gun point to open the safe of Chidamoyo Hospital at 4 AM and surrender the institution’s opulently rich cash reserve of USD $1700 – to a dozen desperately impoverished soldiers of the Zimbabwe National Army (ZNA) who, when caught by police and handed back to the ZNA, promptly found the military’s cover and support (https://www.zimbabwesituation.com/news/12-armed-robbers-arrested-after-robbing- chidamoyo-mission-hospital/ ; http://newsofthesouth.com/new-twist-to-chinhoyi-zna- officers-armed-robbery-case/ ). 

I respectfully submit that the degree of poverty in the authors’ study population is beyond the imagination of the very most readers of PLOS ONE. As is the extent of HIV in the study population; and likewise the extent of tuberculosis in consequence of HIV. Under such conditions, to achieve “suppression increased from 68% in 2016 to 81% in 2018 ” in a population of economically marginalized adolescents in the remote hill country of Northern Zimbabwe is a most extraordinary achievement in the fight against a global vial epidemic; even more so is the data collection and the scientific analysis of the authors - - not just 1 

PONE-D-20-20773-R1 

relevant in the HIV context, but also relevant in the tuberculosis context, an aspect that MUST be introduced in the Discussion (and yes, Chidamoyo Hospital also treats HIV-TB and TB patients). Making pills to fight HIV globally and making vaccines to fight SARS-Cov-2 globally has no effect on these viral pandemics. For effect, we need institutions like Chidamoyo Hospital. For effect, we need nurses, physicians, and scientists exactly like these authors. 

Their voice must be heard. I encourage the authors not to hesitate in making the explicit connection to SARS-Cov-2 in their Discussion – who else is there to administer the SARS-Cov-2 vaccines and to capture the data of that vaccine’s effect but nurses, physicians, and scientists like them? Their manuscript is not just HIV health care delivery, it is implicitly a model of anti-viral health care delivery where it counts most – no one is safe until we all are. The authors should consider that SARS-Cov-2 is anticipated to devastate Zimbabwe (https://www.un.org/africarenewal/news/coronavirus/covid-19-could-prove-“disastrous”- zimbabwe-undp-study-finds ; https://allafrica.com/stories/202010230244.html ; https://allafrica.com/stories/202010080945.html ). They have proven that they can deliver anti-viral care under conditions of extreme poverty, and they should spell it out. A few low-key sentences in the Discussion, with pointers already in the Introduction, are sufficient. 

Response: Thank you very much. The authors share the reviewer’s enthusiasm for all the work done by the Chidamoyo Christian hospital. The authors have added description of the impoverished circumstances in rural Zimbabwe specifically the extreme poverty as described by the National Zimbabwe statistical Center. See highlighted under study setting lines 163-166. 

The authors have added a brief summary of the medical services provided by Chidamoyo Christian Hospital. See highlighted under study setting lines 165-172 and a link to a video presentation on Chidamoyo.

We thank the reviewer for highlighting the findings important to HIV care in a disadvantage population. We have added the sentence “We found that, suppression increased from 68% in 2016 to 81% in 2018 in a population of economically marginalized children and adolescents in an impoverished rural community in Mashonaland West province”. See highlighted in the discussion section lines 329-331. 

In the third paragraph, we have emphasized on the economic and logistic challenges in delivering ART in rural communities. See highlighted lines 349-359. This paragraph points out the challenges of transport, distance and basic infrastructures.

2. The map on the location of the hospital and the area it serves still is far from optimal and not meeting PLOS standards. May I be allowed to offer the authors a graphical proposal (attached). The approximate location of Chidamoyo Hospital is indicated by the white star; the relative locations of the five outpatient sites Batanayi (not ‘Batanai’ !), Magororo, Chedope, Nyamutora, and Zvarai are indicated by white circles as transcribed from the cartographic identifiers provided in https://www.humanitarianresponse.info/sites/www.humanitarianresponse.info/files/ZWE_ MashWest_Province_A0_v1.pdf . 

Response: The authors appreciate the value of the attached figure and have included this rather than the original figure. Thank you very much.

3. Please aim for an abbreviation-minimized reading experience. Remove from the text (not the tables) whenever possible any HIV specialist letter salad and alphabet soup like VL, NNRTI, 3TC, EFV, TDF/3TC/EFV and even TDF +3TC +NVP – please consider typing out the actual names - - you write for a broad audience! Please check again spelling infelicities and like imprecisions. PLOS does not do text editing and proof-reading, that is your responsibility! Please be diligent and consistent to the extreme – it is either “second-line” or “second line”! 

Response: Thank you for this. We have minimized abbreviations throughout the manuscript and be consistent with hyphenation of second-line.

4. In the Introduction the statement of the Results “switched to second-line boosted PI ART with abacavir” clashes with the immediately following statement in the Conclusion “switching to second line ritonavir boosted PI-based ART” – what is it: abacavir or ritonavir? This clash is repeated later in the text. Please address and resolve. 

I suggest one more round of review. Hartmut M. Hanauske-Abel, MD PhD. 

Response: Thank you. The reviewer should note that, second-line protease inhibitor (PI)-based regimens are always “boosted” meaning that they are co-formulated with a low dose of ritonavir. Abacavir and lamivudine are nucleotide reverse transcriptase inhibitors included in second-line regimens. We have resolved this ambiguity in the abstract and throughout the manuscript. See highlighted lines 64 and 67-68.

---

## [Decision Letter · Decision Letter 2]

22 Dec 2020

Viral load care of HIV-1 infected children and adolescents: a longitudinal study in rural Zimbabwe.

PONE-D-20-20773R2

Dear Dr. Mapangisana,

We’re pleased to inform you that your manuscript has been judged scientifically suitable for publication and will be formally accepted for publication once it meets all outstanding technical requirements.

Kind regards,

Joseph Fokam, Ph.D

Academic Editor

PLOS ONE

Additional Editor Comments (optional):

Reviewers' comments:

Reviewer's Responses to Questions

**Comments to the Author**

1. If the authors have adequately addressed your comments raised in a previous round of review and you feel that this manuscript is now acceptable for publication, you may indicate that here to bypass the “Comments to the Author” section, enter your conflict of interest statement in the “Confidential to Editor” section, and submit your "Accept" recommendation.

Reviewer #1: All comments have been addressed

Reviewer #2: All comments have been addressed

2. Is the manuscript technically sound, and do the data support the conclusions?

Reviewer #1: Yes

Reviewer #2: Yes

3. Has the statistical analysis been performed appropriately and rigorously? 

Reviewer #1: Yes

Reviewer #2: Yes

4. Have the authors made all data underlying the findings in their manuscript fully available?

Reviewer #1: Yes

Reviewer #2: Yes

5. Is the manuscript presented in an intelligible fashion and written in standard English?

Reviewer #1: Yes

Reviewer #2: Yes

6. Review Comments to the Author

Reviewer #1: I appreciate the good work, dedication, and responsiveness of the authors.

I respectfully submit for their consideration that, notwithstanding the readability of their manuscript, there still are options to polish the text, e.g. with regard to punctuation. As senior author and experienced 'editorial polisher and polished editor', Prof. Katzenstein might be willing to have a final go at the text and make it shine for eternity.

The authors and PLOS ONE staff should not hesitate to contact me at hanaushm@mac.com in case further graphics work (resolution or pixelation issues, letter sizes and placement etc.) be needed for Figure 1.

I am ready to assist.

Thank you.

H

Reviewer #2: Responses to reviewers is well done. Nice manuscript which should be a good contribution to the field.

7. PLOS authors have the option to publish the peer review history of their article (what does this mean?). If published, this will include your full peer review and any attached files.

Reviewer #1: **Yes: **Hartmut M. Hanauske-Abel, MD PhD

Reviewer #2: No

---

## [Editor Report · Acceptance letter]

6 Jan 2021

PONE-D-20-20773R2 

Viral load care of HIV-1 infected children and adolescents: a longitudinal study in rural Zimbabwe. 

Dear Dr. Mapangisana:

I'm pleased to inform you that your manuscript has been deemed suitable for publication in PLOS ONE. Congratulations! Your manuscript is now with our production department. 

Kind regards, 

on behalf of

Dr. Joseph Fokam 

Academic Editor

PLOS ONE